# Chiral Recognition for Chromatography and Membrane-Based Separations: Recent Developments and Future Prospects

**DOI:** 10.3390/molecules26041145

**Published:** 2021-02-21

**Authors:** Yuan Zhao, Xuecheng Zhu, Wei Jiang, Huilin Liu, Baoguo Sun

**Affiliations:** Beijing Advanced Innovation Center for Food Nutrition and Human Health, Beijing Technology and Business University, 11 Fucheng Road, Beijing 100048, China; 15624986601@163.com (Y.Z.); zhu_xuecheng@163.com (X.Z.); 18811025200@163.com (W.J.); sunbg@btbu.edu.cn (B.S.)

**Keywords:** chromatography, membrane, chiral stationary phases, chiral separation, recognition

## Abstract

With the rapid development of global industry and increasingly frequent product circulation, the separation and detection of chiral drugs/pesticides are becoming increasingly important. The chiral nature of substances can result in harm to the human body, and the selective endocrine-disrupting effect of drug enantiomers is caused by differential enantiospecific binding to receptors. This review is devoted to the specific recognition and resolution of chiral molecules by chromatography and membrane-based enantioseparation techniques. Chromatographic enantiomer separations with chiral stationary phase (CSP)-based columns and membrane-based enantiomer filtration are detailed. In addition, the unique properties of these chiral resolution methods have been summarized for practical applications in the chemistry, environment, biology, medicine, and food industries. We further discussed the recognition mechanism in analytical enantioseparations and analyzed recent developments and future prospects of chromatographic and membrane-based enantioseparations.

## 1. Introduction

Enantiomers are defined as a pair of compounds that are non-superimposable mirror images of each other, and such compounds are usually called chiral molecules. They were first proposed by Louis Pasteur in 1848 [1] when he used a magnifying glass to manually separate two enantiomers in a crystal of ammonium tartrate. The in-depth study of enantiomers has shown that the configurations of two enantiomers in space are not identical, having a similar relationship as the left and right hands. Chiral molecules, means that a molecule whose configuration does not coincide with its mirrors [2,3]. Most of the molecules at the center of life, including DNA, amino acids (AAs), and sugars, are chiral [4,5]. The resolution and analysis of chiral compounds is not only a key link in scientific research but also a potential application in pharmaceutical analysis, food, environmental, clinical, synthetic and analytical chemistry.

Chiral separation is similar to chemical separation, which is the opposite of the mixing process driven by the second law of thermodynamics because the chiral separation process is normally not a spontaneous procedure. In recent years, more and more studies have shown that enantiomers have different pharmacological activities. These enantiomers may affect human health and ecological stability. For example, the endocrine disrupting effects of chiral pesticide enantiomers on organisms are quite different. In recent years, an increasing number of studies have shown that chiral substances are harmful to the human body based on their chiral properties, and the selective endocrine-disrupting effect of enantiomers of chiral drugs is caused by the different affinities in enantiospecific binding to receptors [6]. With the rapid development of global industry and increasingly frequent product circulation, the separation and detection of chiral drugs/pesticides are becoming increasingly important. In addition, chiral pesticides not only affect the quality of food through residues during spraying but also affect aquatic life by flowing into the water environment after spraying, thereby affecting aquatic product quality and even more seriously producing lethal toxicity to aquatic life [7]. Generally, enantiomeric ratio (ER) or enantiomeric fraction (EF) is used as an index for evaluating the ratio of enantiomers. The calculation formulas of ER and EF are shown in Formulas (1) and (2). Under normal circumstances, the racemate needs ER = 1, EF = 0.5, deviation from this value indicates the enantiomeric excess [8].
(1)$$\mathrm{ER}\;=\;\frac{\mathrm{Right}-\mathrm{handed}\;\mathrm{molecule}\;\left(+\right)}{\mathrm{Left}-\mathrm{handed}\;\mathrm{molecule}\;\left(-\right)}$$
(2)$$\mathrm{EF}\;=\;\frac{\mathrm{ER}}{\mathrm{ER}+1}$$

Chiral separation techniques include CSP-based chromatographic separation (e.g., liquid chromatography (LC) (including high-performance liquid chromatography (HPLC), hydrophilic interaction chromatography (HILIC), and ion exchange chromatography (IEC)), gas chromatography (GC), capillary electrophoresis (CE), supercritical fluid chromatography (SFC), thin-layer chromatography (TLC), and high-speed countercurrent chromatography (HSCCC) and membrane-based separation (e.g., liquid and solid membrane separation). The rapidly increasing amount of research activity related to chiral selectors (CSs) for separation applications in the field of chromatography and membrane-based separations is shown by the plot in Figure 1.

In this review, we focus on recent advances in chiral separation, including versatile perspectives on current advantages, bottlenecks and prospects for the future development of chromatographic and membrane-based separations.

## 2. Chromatographic Separation

Chromatographic resolution is a chiral resolution method that has developed rapidly in recent years. The mobile phase and CSP-based chromatographic column are the key factors for chiral chromatographic separation. Based on this, the chiral compound is dissolved in the solvent used as the mobile phase, due to the different adsorption, distribution and affinity of the stationary phase with respect to chiral substances. The mobile phase will have a different residence time flowing through a CSP-based chromatographic column modified with CS, such as a chiral molecule or a molecule modified with a chiral surface structure for the separation of chiral compounds. According to the different response time of CSPs, each liquid part was collected to obtain different chiral compound. [9]. In addition, the thermodynamic parameters also indicate that the separation of enantiomers on CSPs is caused by differences in the forces between the enantiomers, which are mainly influenced by hydrogen π-π interactions, induction, dispersion, and orientation interactions [10,11]. At present, many types of CSPs have been successfully commercialized, and they are combined with chromatography technology to achieve the separation of enantiomers. Table 1 classifies the scope of application of different CSPs and commercialized chromatographic columns.

### 2.1. Liquid Chromatography

HPLC is a fast, specific, and highly efficient method using different molecular sizes, spatial structures, and affinities to effect separation. After separation, detectors can accurately detect the properties of the separated molecules. Its advantages are good selectivity, high sensitivity, and wide application range. Its disadvantage is that it requires a large amount of organic solvent as the mobile phase. At present, liquid chromatography has developed a variety of stationary phases derived from small molecules, polysaccharides, macrocyclic antibiotics, and chiral crown ethers for chiral separation, which are used for the resolution of enantiomers. The chiral recognition mechanism mostly relies on empirical development, and some targeted special designs are still constantly updated [12].

Polysaccharides that are highly effective CSs may include phenyl, alkyl or benzyl carbamate, ester, benzoate, aryl, or cycloalkyl groups, thus giving the CSP different chiral recognition capabilities [13]. Among a wide range of polysaccharide derivatives, 3,5-dimethylphenyl derivatives have been proven to have the best chiral recognition ability and are used as excellent materials in optimization experiments [14,15]. In addition, cellulose is called microcrystalline cellulose triacetate. Because it has a regular three-dimensional structure and can be easily modified into esters and urethanes, it is suitable as CSPs. When they are combined, they form cellulose tris (3,5-dimethylphenyl carbamate) with powerful and extensive chiral recognition ability, so it is finally used for commercialization and named Chiralcel OD [16]. Polysaccharides are not only useful for resolving enantiomers, but recent studies have shown their potential value in resolving non-enantiomer mixtures. This is especially true for separating isomers with double bonds [17].

Chitin and chitosan are difficult to dissolve in common organic solvents, and they also have some chiral separation capabilities, so they have also been developed in recent years [18]. The main advantages of chitin and chitosan for CSPs are their reproducibility and good mobile phase tolerance [19,20,21]. Using renewable chitin as raw material, Nguyen [19] has developed a simple, environmentally mild and sacrificial-template-free method for preparing *N*-doped chiral nematic carbon sheath nanofibril hydrogel with specific surface area >300 m^2^g^−1^ and good enantioselectivity. It has different chiral adsorption capacity for l-lactic acid and d-lactic acid (Figure 2a). Feng and colleagues conducted experiments on the effect of chiral separation of chitosan with different molecular weights. The synthesized materials CSP3 and CSP5 derived from chitosan bis(arylcarbamate)-(amide)s are significantly better separators than cellulose tris(3,5-dimethylphenylcarbamate), under the condition of n-hexane: Isopropanol = 1:1 (*v*/*v*) with the flow rate of 1 mL/min. Among the tested enantiomers, the total number of chiral compounds that can be completely separated by chitosan derivatives is higher than that of cellulose tris(3,5-dimethylphenyl carbamate). Through research, it is found that the CSPs with CH_3_-π bonds have a lower swelling and higher stability because they diminish the role of the solvent. The degree of swelling is used to reflect tolerance to the mobile phase. In addition, hydrogen bonding is also an important force affecting mobile phase tolerance. Low molecular weight chitosan bis(3,5-dichlorophenylcarbamate)n-pentanamide-derived CSPs have good enantiomeric resolution, but the mobile phase of these CSPs is not as tolerant as glycan bis(3,5-dichlorophenylcarbamate)-(n-valeramide), so when selecting chitin, its chiral separation ability and mobile phase tolerance should be balanced [22].

Cyclofructan (CF) is usually composed of six or more linked fructofuranose units. The most commonly used cyclofructan for chiral recognition is cyclofructan-6, which consists of six d-fructofuranose units (Figure 2b). In 2009, cyclofructan was first used in an experiment for enantiomeric separation of bonded CSPs. The chiral recognition ability of natural cyclofructan is limited, but the derivatized cyclofructan has been proven to have better chiral recognition ability [23]. Among them, when cyclofructan is partially derivatized with aliphatic functional groups, it shows a better separation effect on primary amines. This is because when its hydroxyl group is partially derivatized, the hydrogen bond is inhibited, resulting in exposure of the crown ether center. This functionalized crown ether has better selectivity to the primary amine enantiomer [24]. Highly aromatic functionalized cyclofructans lose their enantioselectivity to primary amines, but they have broad selectivity for most other types of analytes. For example, this process can obtain a selection factor of 1.91 and a resolution (Rs) of 4.4 when *N*-(3,5-dinitrobenzoyl)-dl-leucine is separated [25]. Rs is a parameter for judging the separation of substances to be separated in the chromatographic column. Its calculation formula is shown in Formula (3). Among them, *n* is the average effective plate number, α is the separation factor, *k* is the average capacity factor of the two components, and k’ is the capacity factor of one of the components. Generally, Rs ≥ 1.5 proves complete separation.
(3)$$\mathrm{Rs}\;=\;\frac{n^{\frac{1}{2}}}{4}\;\times \;\frac{\alpha -1}{\alpha }\;\times \;\frac{k^{\prime }}{k\;+\;1}$$

Commonly used commercially available cyclofructan columns are isopropyl carbamate cyclofructan-6, *R*-naphthylethyl-cyclofructan-6 and dimethylphenyl-cyclofructan-7, and these cyclofructose columns have been shown to be useful for the separation of multiple enantiomers. Compared with polysaccharide-derived CSPs, these have a lower operating temperature, a higher mobile phase flow rate, and a wider range of mobile phase types [24,26,27]. Regarding the expansion of the application field of cyclofructan CSPs and the improvement of their chiral recognition ability, the main directions in recent years have been the diversification of substituents during derivatization and the application of superficially porous particles. Khan and his colleagues used cyclofructan-6 to prepare a series of chloroaromatic derivatives for the preparation of CSPs. The results show that among the 10 new CSPs synthesized, the chloromethyl phenyl derivatives were substituted at the 3, 4 or 4, 3 positions. Their enantiomeric separations were strongest among these 10 CSPs. Their enantiomeric separation abilities were almost the same as those of the cyclofructan CSPs on the market, but the synthesis of the new CSP will provide a basis for the expansion of the application field in the future [28]. Breitbach used superficially porous particle columns for research to improve column efficiency and used this column for ultra-high performance liquid chromatography to improve separation efficiency [25].

CD is a class of cyclic oligosaccharides composed of α-1,4-linked d-glucose molecules. In chiral material separation, natural CD also has some limitations. For example, its low water solubility will cause the optimal CD concentration required for chiral identification to exceed its own solubility limit [29,30]. The addition of new groups can improve the enantiomeric separation ability of CDs. Li et al. reported a new *N*-benzyl-phenethylamino-β-CD-bonded CSP prepared by introducing *N*-benzyl-phenethylamino that can be used in various elution modes. Under polar organic elution modes, it effectively separated nine β-adrenergic blocker drugs within 20 min, where the Rs of carvedilol was 1.97. The main forces in this mode are inclusion and hydrogen bonding. In reversed-phase (RP) elution modes, the Rs of eight dansyl AAs (DNS-AAs) and six flavanones showed that this material efficiently resolves enantiomers. It can successfully separate the dansyl-tyrosine enantiomer in 20 min, and the obtained Rs was 3.29. It can also successfully separate the 4′-hydroxy flavanone enantiomer in 15 min at a Rs of 3.65. The main forces in this mode are inclusion and hydrophobic interactions. This new CSP has a faster separation speed and higher separation selectivity than commercially available CD CSPs. The above results indicate that *N*-benzyl-phenethylamino participates in the chiral separation and enhances the separation efficacy [31]. Sun et al. prepared fully derivatized 4-chlorophenylcarbamate-β-CD bonded CSPs. In enantiomeric separation experiments on 22 drugs, all the enantiomers were successfully separated. Among them, the best separation effect was for azoles (Rs = 16.80) compared to the separations by commercial 3,5-dimethylphenyl carbamate-β-CD-based CSPs, but these materials exhibited improved separation of all enantiomers. The above results indicate that the 4-chlorophenylcarbamate group participates in the chiral separation process and enhances the separation efficacy. Further research proved that the enhanced forces are inclusion complexation, hydrogen bonding, π-π stacking interactions, dipole–dipole interactions and steric hindrance [32]. Zhou et al. prepared novel bilayer cationic CD CSPs through a click reaction and tested them in RP and normal-phase (NP) elution modes (Figure 2c). Seventeen enantiomers were tested, and organic modifiers were added to the mobile phase to improve the separation ability. Twelve species achieved high-efficiency separation in RP elution mode. The Rs of 4-(chlorophenyl) propyl ester reached 4.40. By bonding the regioisomer cationic cyclodextrin outside the natural cyclodextrin inclusion compound, the additional interactions are increased, thereby enhancing the ability of chiral recognition. The click reaction can realize the construction of bilayer CD-CSP [33]. Li also reported this bilayer CD-CSP, and achieved complete separation of dansyl-dl-leucine and dansyl-dl-phenylalanine within 30 min [34].

In addition, the preparation methods of CSPs are very important for the chiral separation. Typically, click chemistry is widely used in the synthesis of CSPs because it has a great degree of choice of solvent, superior chemoselectivity and mild reaction conditions [35]. Zhou and his colleagues linked perphenylcarbamate β-CD chloride to an alkynyl silica via a click reaction. Performing resolution experiments on 21 racemates showed that the introduction of a 3-methoxypropylammonium moiety on CD CSPs can achieve additional separation efficacy. The Rs of 7-methoxyflavonoids in NP mode is 9.84. In the evaluation of the chiral separation experiment of 7 kinds of flavonoids, it was concluded that the type and position of functional groups in the flavonoid compounds play a key role in their chiral separation [36]. In another experiment, the click reaction was again used to synthesize four perphenylcarbamate CD clicked CSPs. Through analysis, they found that per(3-chloro-4-methylphenylcarbamate) CD clicked CSPs showed the best chiral separation ability. In a ternary elution system, a Rs of 20 was achieved for 4ClPh-OPr. Comparing the four CSPs showed that in addition to the type of CSPs, the position of the substituents largely determines the chiral separation ability [37]. In addition, there have been many experiments that use click chemistry in the preparation of CSPs [38]. A similar bridging linker structure which using click chemistry was reported for bridged bis (β-CD) by Shuang et al. These results further proved that the synergistic inclusion ability of the bridging linker and two adjacent cavities is the key to improving chiral recognition [39,40].

A CD CSP based on graphene oxide was developed for HILIC (Figure 3a,b), and the study found that graphene oxide and β-CD play a synergistic role in the recognition process [41]. HILIC is generally used to analyze highly polar compounds. Compared with traditional normal phase chromatography, the sample solvent can contain a certain proportion of water when using HILIC, which can dissolve some hydrophilic compounds that are hard to dissolve in organic solvents. Tang et al. synthesized a CSP using a light-assisted strategy. Diazo resin and CD-COOH were bound to silica particles by ionic bonds; Diazo resin, CD-COOH, and silica were coupled by UV light; and the ionic bonds were converted into covalent bonds. These CSPs can be obtained by enantiomeric separation experiments to separate analytes. Promethazine and benzoin enantiomers were separated in polar organic elution modes (Rs are 2.675 and 1.131, respectively). This method used water-soluble and nontoxic Diazo resin instead of the highly toxic silane agent to modify silica microspheres, thereby realizing the synthesis of CSPs in a green and effective way (Figure 3c) [42].

Ion-exchange CSPs are generally divided into three categories: Chiral weak anion-exchange CSPs, chiral strong cation-exchange CSPs and zwitterionic ion-exchange CSPs. IEC is mainly used to separate ionizable compounds, such as peptides, proteins, and nucleotides [43]. In 1985, Rosini et al. used cinchona alkaloids for the first time to prepare ion-exchange CSPs [44]. Later, research on ion-exchange CSPs continued, but the chiral separation ability was generally low until Lindner et al. proposed a simple structural modification method in the preparation of ion-exchange CSPs. By using different types of spacers and grafting modes to immobilize quinine and quinidine carbamates on porous silica, a series of acidic enantiomers were successfully resolved [45]. A series of commercial ion-exchange-type CSPs with efficient separation capabilities exist today (Figure 3d) [43]. Anion-exchange stationary phases are usually formed based on cinchona alkaloids. The chiral C8 and C9 atoms of the cinchona backbone are the key to chiral spatial recognition. A cation-exchange phases are formed based on sulfonic acid or carboxylic acid groups. Zwitterionic ion-exchange CSPs are a combination of cinchona alkaloid units and sulfonic acid or carboxylic acid units. Compared with the previous two types of CSPs, this type of CSP has a greatly broadened scope of application and is not limited to identifying singly-charged enantiomers [10]. Zwitterionic ion-exchange CSPs were first studied by Lindner and his colleagues and are currently mainly used for enantiomeric recognition of α-, β-, and γ-AAs and short peptides. Polar organic elution and RP elution are mainly used [46,47,48,49].

The stability of a material is as important as chiral separation for CSPs. Common methods to improve stability are coating CSPs and covalently immobilizing CSPs [13,50]. Generally, immobilized CSPs are more stable than coated CSPs because the chemical groups introduced by bonding cause partial changes in the regular spatial structure of the polysaccharides; the former CSPs has a lower chiral recognition ability than the latter [51]. In addition, silica is considered a good column carrier in HPLC, the particle diameter of the analytical column is generally 3–7 μm, and the particle diameter in the prepared column is 5–20 μm. Compared with silica, general inert carriers, such as titanium oxide, zirconia, magnesium oxide, and titanate silica, have no obvious advantages. In addition to the widely used silicon-based monolithic columns, the latest improvement method now available is the use of superficially porous silica, optimization of its pore size, and shell coating with porous silica as the core. Porous silica is used as the core and silver nanoparticles as the shell in the formation of SiO_2_@Ag material, and CSPs prepared with a cellulose derivative as a CS were compared to CSPs without silver nanoparticle modification of recognition. After modification, the separation effect obtained by chromatographic analysis of the racemates of 15 enantiomers was improved [52,53].

### 2.2. Gas Chromatography

In the 1960s, Gil-Av and Feibush were the first to separate enantiomers by GC [54]. Under GC conditions, chiral separation is mainly performed on a CSP capable of hydrogen bond coordination and inclusion. Commonly used are amino acid derivatives, terpene-derived metal coordination compounds and modified CD. It appears that most enantiomers are separated using CD [55]. There are two basic types of monolithic columns modified with CD: Monolithic silicon columns prepared by a polycondensation reaction of an alkoxysilane using a sol-gel technique and monoliths obtained by polymerization in situ. Polymer chromatographic columns are prepared in the presence of a pore-forming solvent. The second monolithic chiral column is more widely used in GC. In this case, a polymer obtained by a well-known monomer polymerization method is used for the preparation of a monolithic column, and CD is covalently bonded to its inner surface [56]. In 1982, Smolková et al. started using CD to separate enantiomers [57]. At present, a modified CD chemically bonded to a functional group on a siloxane carrier has overcome the problem that pure CD cannot be applied to GC due to its high polarity. The β-CD derivatives are currently used as the main CD-based CSPs for the separation of various enantiomers (Figure 4a) [58,59]. Water-soluble polymers (polyethylene glycol, sodium carboxymethyl cellulose, polyvinylpyrrolidone, etc.) have been widely used to improve the binding efficiency of enantiomers with CD [60]. Due to the advantages of low solubility, cost and toxicity of polyethylene glycol in water, researchers built organic compound and β-CD models in polyethylene glycol solution to separation enantiomer [61]. Compared with HPLC, GC could not only analyze samples with good thermal stability and volatile, but also with lower detection limit. Because the mobile phase of GC is gas, it will not produce a large amount of organic waste liquid.

However, the resolution capability of a single CD derivative-based capillary column is limited, and commercialized cyclodextrin derivative capillary columns are expensive. Therefore, it is important to develop CSPs with higher enantioselectivity for GC. Here we mainly introduce the rapidly developing chiral porous materials and new supramolecular structures CSPs. Chiral porous materials include covalent organic frameworks (COFs), metal organic frameworks (MOFs), porous organic cages (POCs), metal organic cages (MOCs), and chiral mesoporous silicas (CMS).

Due to the difficulty of directly designing and synthesizing MOFs with chiral recognition sites, Kou et al. used post-synthesis methods to prepare chiral MOFs for gas chromatography. Five chiral ligands were selected, and five chiral MOFs were synthesized by grafting different ligands on MIL-101-NH2. They are mainly recognized by hydrogen bonds and π-π interactions, so they combine the advantages of large specific surface area and strong adsorption affinity of MOFs [62].

COFs is made up of organic structural units connected by strong covalent bonds. It is a new type of material with low density, high stability, and large specific surface area. Chiral COFs is difficult to synthesize, so the current general method of constructing COFs with chiral is to introduce chiral functional groups through bottom-up strategy. Qian et al. prepared a chiral functional COF by a bottom-up strategy, and prepared a chiral COF capillary column by the in-situ growth approach. Using it to test (±)-1-phenylethanol, (±)-1-phenyl-1-propanol, (±)-limonene, and (±)-methyl lactate, it is concluded that all four enantiomers can be Separate on GC within 6 min [63].

Unlike COFs and MOFs, POCs, and MOCs form an ordered porous structure through weak interactions between molecules. Therefore, their structure is looser than COFs and MOFs. Their advantage is that they can be dissolved in common organic solvents, and they are easy to change the structure and make composite materials. Zhang et al. used a homochiral porous organic cage (CC3-R) diluted with polysiloxane (OV-1701) as the stationary phase of GC to achieve the separation of multiple enantiomers (Figure 4b). Compared with commercial β-DEX120 and Chirasil-l-Val columns, CC3-R capillary columns have better separation capabilities [64]. Sheng et al. prepared a homochiral MOC[Zn_3_L_2_] coated capillary column and used it to achieve the separation of multiple enantiomers. Compared with the commercial β-DEX 120 column and homochiral porous organic cage CC3-R coated column, it is found that it has better separation ability [65].

CMS material was discovered in 1992. It has the advantages of large specific surface area, adjustable structure and high temperature resistance, so it is suitable for GC. He et al. synthesized CMS by self-assembly of the achiral surfactant sodium dodecyl sulfate in the presence of chiral amino alcohol, and then prepared a CMS coated column. It has excellent chiral recognition ability for chiral alcohols, ketones, aldehydes, organic acids, olefins, halogenated hydrocarbons, amino alcohols, epoxides, and amino acid derivatives [66].

At present, there are many studies analyzing localized chiral regions in supramolecular structures of nonchiral molecules by scanning tunneling microscopy and density functional theory methods [67,68]. A chiral phase based on nanomaterials has better enantioselectivity and stability than CD chiral phases [69,70,71], and a two-dimensional supramolecular structure of heterocyclic compounds is distinguishable in nanomaterials. This structure can be easily self-assembled on virtually any surface and can form remote surfaces with long-range order. Chirality can be induced to form superstructures using various external factors even if the original monomer has no chirality [72].

Based on the self-assembled supramolecular structure of melamine, the team of Gainullina [73] proposed a new CSP that was generated by chiral induction. It can be used for GC. According to the Kondepudi effect [74], a supramolecular layer composed mainly of a class of chiral supramolecular domains was formed on the surface of an adsorbent by mechanical stirring and hybridization on the surface of an inert solid carrier. This layer can be used for the separation of 2-butanol, 2-bromobutane, and other substances by GC, in which the cyclization selectivity is the result of the removal of the central molecule of the supramolecular structure at 200 °C. Similarly, in 2019, Nafikova et al. [75] investigated a uracil structure as a stationary phase and successfully separated the enantiomers of several substances. In the temperature range studied, the enantiomers of 2-bromobutane and 2-chlorobutane were completely separated at 45 °C in 210 and 180 s, and the enantiomers of 2-chlorobutane were separated at 60 °C in 160 s. Guskov et al. [74] used indirect methods of static adsorption with polarization control and GC to differentially adsorb enantiomers on modified adsorbents, demonstrating the usefulness of chiral recognition scaffolds (Figure 4c). The stability of the supramolecular structure and how to effectively construct the required molecular domain are still problems.

### 2.3. Capillary Electrochromatography

Since capillary electrochromatography (CEC) was first applied to chiral separation in 1985, CSs have become a research hotspot, especially the introduction of the first CE instrument in 1988, which has promoted the development of this field. The basis for separation using CSs is the formation of host-guest complexes through noncovalent bonding interactions (i.e., hydrogen bonding, hydrophobic forces, electrostatic forces, and steric hindrance effects) [76]. When using this method for enantiomeric separation, the chiral selection agent needs to be added to only the background electrolyte to provide greater flexibility, selectivity, and resolution in the optimization of separation [10,77,78].

Ionic liquids (ILs), also known as room-temperature ILs, are salts composed of organic cations and inorganic/organic anions that are liquid at or near room temperature [79,80]. ILs have the advantages of high solubility and high conductivity in water [81,82,83,84] and have been widely used in CEC technology. The earliest use of ILs in analytical science dates back to 1966. Scientists tried to extract metal ions with quaternary ammonium halide ILs and organic solvents, but it did not receive much attention until the emergence of hydrophobic anionic ILs in the 1990s. Research in fluids has flourished. According to current literature reports, Xu et al. [85] synthesized a chiral IL, cholinium-clindamycin phosphate (Ch-CP), as a single CS in CE, highlighting the performance of a single IL as a potential CS.

In addition to being used in HPLC and GC, CD is also an important CS in CEC. Some CD derivatives with strong water solubility and high electrophoretic mobility have been continuously developed and applied. For example, uncharged CD derivatives include methylated, hydroxypropylated, and acetylated CDs, positively-charged CD derivatives, that is, various amine-modified CDs, include amphoteric-β-CD (AM-β-CD), 6-amino-2-deoxy-β-CD (ACD), mono-(6-β-aminoethylamine-6-deoxy)-β-CD, and 6-deoxy-6-hexylamino-β-CD (HACD), and negatively-charged CD derivatives include sulfonated, carboxymethylated, and sulfonated butylated CD [86]. Gábor Benkovics [87] and his team synthesized a new monoisomer carboxymethyl-γ-CD, octakis-(2,3-di-O-methyl-6-O-carboxymethyl)-γ-CD sodium salt (ODMCM), used racemic dapoxetine as an analyte, and analysed the interaction between host and guest at pH 2.5 and pH 7.0 at the molecular level by nuclear magnetic resonance (NMR) spectroscopy. Under ^1^H NMR spectroscopy, the chemical shift for a single chiral substance caused by complexation can be observed at two pH values, and the S-dopastatin enantiomer and the standard dapoxetine-ODMCM system were used simultaneously to demonstrate ^1^H-NMR resonance shifts. When the spectra were compared at two pH values, it was found that there was a more pronounced complexation at pH 7.0 for the new material (ODMCM and dapoxetine were in the forms of polyanions and monocations, respectively). Thus, the new material exhibited better enantioselective recognition of the dapoxetine isomer. In another study [88], a novel long-chain receptor was synthesized by bridging a trehalose capping unit with the interior of the CD cavity through lysine residues and then characterized. Lysine-bridged hemisphere dextrin, which is a β-CD-terminated derivative, showed that this kind of derivative can achieve good separation even as a CS in CE alone. With the progress of separation technology, more research has focused on the use of a dual selection system containing CD to better improve separation.

Macrocyclic antibiotics can be chiral selection agents because they were combined with a d-alanine-d-alanine (d-Ala-d-Ala) terminal group to achieve antibacterial effects [89]. Macrocyclic glycopeptide CSPs have become the first choice for enantiomeric separation of natural AAs and amino acid derivatives, especially for the enantiomeric separation of β-AAs [90,91]. We know that AAs are the basic units of peptides. Peptides can be chemically modified to increase their utilization, thereby preventing them from being broken down by human proteases and peptidases. As a result, the number of peptide drugs and nutritional products is increasing year by year [92]. Structurally, many drugs are related in their structure: Because the spatial configuration of an AA is different (chiral diversity), its effects are also different [93]. Therefore, the identification of chiral AAs with high efficiency and high accuracy is an important issue. In most cases, macrocyclic antibiotics have high stereoselectivity for compounds containing acidic groups and anions, such as carboxylates, phosphates, and sulfates. Based on this situation, Zhang [94] and colleagues chose streptomycin as a CS for the separation of five acidic drug enantiomers. Streptomycin had weak absorption at ultraviolet wavelengths when chiral separation was successfully achieved after CE through uncoated capillaries. Dixit and others [95] first applied rifampicin as a CS in CE separation. Rifampicin contains nine stereoisomeric centers, an aromatic group and different functional groups, including imine, amide, acetoxy, and aliphatic hydroxyl groups, which can interact with the expected chiral compounds to achieve separation.

Chiral crown ethers are excellent CSs in CE. Racemic AAs and protonated primary amines have a high degree of complexation selectivity because protonated primary amines and ammonium are bound by hydrogen bonds arranged on three legs of a tripod. The structural cavity and cavity size of these crown ethers can just match the shape of the primary amine group, so inclusion is one of the driving forces for chiral resolution. In addition, a carboxy substituent is on the crown ether and the chiral object. The steric hindrance and hydrophobic interactions between the two substituents are also essential driving forces for binding. However, due to the weak ultraviolet absorption of amines and the toxicity of crown ether, the application of these materials is still limited.

To be included in a crown ether, the amino group must be in a protonated state. Therefore, reactions with it are usually carried out at low pH, where the electroosmotic flow is small and the separation time is long [96]. Based on this situation, Hagele [97] and others used (+)-18-crown-6-tetracarboxylic acid (18-C-6-TA) as a chiral selection agent in CE separation and separated fifteen new types of novel compounds in a short time. In addition, 18-C-6-TA can be used in CE as a single chiral selection agent, and it can also be used to form a chiral selection system with other substances for the separation of chiral compounds. Researchers such as Paik [98] dissolved heptakis (2,6-di-*O*-methyl)-β-CD (DM-β-CD) in 18 mM phosphate-triethanolamine buffer (pH 3) in the presence of DM-β-CD solution using 18-C-6-TA as a chiral co-selection agent for chiral resolution of eight analytes. Some other types of CSs are also used in CE chiral separation. For example, Luma et al. [99] reported the mechanism of chiral separation between DNA oligonucleotides (ONs) and low-affinity DNA compounds when CSs were used in CE. The sequence prerequisites of ONs during enantiomeric separation laid the foundation for the design of high-performance ON CSs and CE isolation of weakly acidic DNA chiral compounds. Sara et al. [100] synthesized chiral carbosilane dendrimers using *N*-acetylcysteine groups and cysteine and used these dendrimers as CSs in CE for chiral compounds. It was found that when using four terminal *N*-acetyl-l-cysteine groups, razoxane can be effectively chirally recognized, and its discrimination ability is similar to that of strong CSs such as CD. In addition, capillary zone electrophoresis (CZE), as a separation mode of CE, is widely used in chiral resolution. In addition to the use of aqueous solvents for chiral separation in CZE, non-aqueous capillary electrophoresis (NACE) has also been successfully applied to chiral material analysis. For example, An et al. [101] used lactobionic acid/d-(+)-xylose–boric acid complexes as CSs in chiral NACE. A buffer solution of methanol and triethylamine was used as the background electrolyte and achieved good resolution of fourteen amino alcohol chiral compounds. At the same time, the method was suitable for routine analysis of propranolol enantiomers, and the recovery rate of each propranolol enantiomer was 96.4–105.9%. In recent years, there have been reports about the application of polymers in chiral separation by CE. For example, Bao [102] and others synthesized sulfated cyclodextrins (SCDs) by polycondensation. New sulfoether ether γ-CD polymer (SPE-γ-CDP). The polymer has high viscosity and chiral selectivity. After the polymer was characterized by infrared (IR) spectroscopy and indirect UV detection, the best separation was achieved. Under these conditions, it was successfully used as a CS in the chiral separation of analytes in CE. Elahe [103] reported a study with a 25 °C uncoated fused silica capillary using 100 mM phosphate buffer (pH 8.0) as the background electrolyte and maltodextrin as a CS. At the same time, baseline resolution of tramadol (TRA) and methadone (MET) chiral compounds was successfully obtained in less than 12 min by applying a potential of 16 kV. Liu et al. [104] used streptomycin-modified gold nanoparticles (ST-AuNPs) as CSs. Generally, the effect of CSs and enantiomers is proportional to the concentration of additives, but in this study, it was found that ST-AuNPs can act as a “selection carrier” and that their own concentration can be used to determine the effect during enantiomer transport. With the decreasing of the concentration of ST-AuNPs, the absorbed enantiomers decreased, and the peak area of the first enantiomer was larger than that of the second enantiomer (Figure 5(a1)). As the concentration of ST-AuNPs increased, the number of chiral compounds it carried increased, and the peak areas of the two enantiomers were almost equal (Figure 5(a2)). Conversely, if the concentration of ST-AuNPs was too large, some selectivity was lost, so that the poorly paired enantiomers were transported at the same time, and the first peak area was smaller than the second peak area (Figure 5(a3)). Therefore, after the optimal concentrations of ST-AuNPs were determined, epinephrine and norepinephrine were tested in uncoated fused silica capillaries. The study found that under optimal conditions, a baseline separation of analytes was achieved within seven minutes, with a maximum Rs of 7.5. Zhang et al. [105] studied the potential of chondroitin sulfate D (CSD) and chondroitin sulfate E (CSE), i.e., sulfated polysaccharides, as CSs in CE separation. Under the optimal conditions, CSE separated chiral compounds more successfully, and CSD enabled partial resolution of the analytes. It was observed that CSE had better chiral recognition of the test compounds than CSD. In recent years [106], *N*-(4-*H*-1,2,4-triazolium)-lactobionamides (LA-ATM) were synthesized for the first time and used in CE for enantiomeric separation. Compared with the simple lactobionamide (LA) system, this system exhibited an increased degree of separation, and the chiral recognition mechanism of LA-ATM was confirmed by molecular modelling.

In addition, covalent organic frameworks (COFs) are also used in CEC due to their excellent properties. Among them, COF LZU1, with good size selectivity and hydrophobic interaction, effects high resolution separation. There is research reporting the use of epitaxial growth to fabricate an in situ COF LZU1-coated capillary column for CEC (Figure 5b) [107]. The baseline separation of neutral analytes, AAs and nonsteroidal anti-inflammatory drugs (NSAIDs) was successfully achieved. Hydrophobic, π-π and hydrogen bonding interactions played a key role in the separation. Recently, Song [108] and colleagues used methacrylic acid as a functional monomer and octavinyl-modified polyhedral oligomeric silsesquioxane (OvPOSS) as a crosslinking agent to obtain molecularly imprinted polymers (MIPs) using S-amlodipine as a template. OvPOSS is easy to prepare, and unlike conventional MIP, when coated on the capillary wall and tested for CEC separation of amlodipine enantiomers, quite good efficiency was obtained (Figure 5c).

In addition, nanoparticles have a unique chemical surface, which can form a stable suspension in the background electrolyte. The combination with polymers can improve the selectivity and repeatability of CE chiral analysis, and gradually become the most promising chiral selector. Related application examples have also been mentioned in previous reviews [109].

### 2.4. Supercritical Fluid Chromatography

SFC combined with effective CSPs, it can be used for enantiomer resolution in pharmaceutical, environmental and food control. Among macrocyclic antibiotics, Pirkle type CSPs, polysaccharide-based columns (cellulose and amylose derivatives) have become the best choice for chiral separation of SFC due to their high loading capacity [110]. They have been shown to allow stereoselective separation of multiple compounds. The same polysaccharide-based CSPs can be used for HPLC and SFC, but they have better selectivity for SFC, a finding attributed to the effect of mobile phase co-solvents. With its high enantioselectivity and resolution, SFC separates enantiomers faster than HPLC. At the same time, as the polarity of the co-solvent (modifier) in the mobile phase was increased, analytes eluted faster. The traditional liquid mobile phase is replaced by dense compressed gas in SFC. Carbon dioxide is the most common supercritical mobile phase. The use of CO_2_ replaces organic reagents is safer and greener [111]. The lower viscosity and higher diffusivity of CO_2_ provide the advantages of higher flow rates, resulting in shorter analysis times and faster column equilibration times. Analyte retention and selectivity largely depend on the density of the mobile phase, which in turn depends on temperature, pressure, and the composition of the mobile phase. Therefore, for different compounds, the chromatographic conditions to achieve acceptable separation results may vary greatly, which makes the optimization of the SFC chiral separation method a more challenging task.

The interaction mechanism of soluble proteins with tris-(3,5-dimethylphenylcarbamate in the liquid phase and the supercritical fluid separation system has been discussed [112]. By comparing the separation capabilities of chiral sulfoxide with different separation technologies, the superior separation effect of SFC and ordinary liquid chromatography (LC) on polysaccharide-derived CSPs was reported, and similar experimental conclusions were obtained [113]. Chen et al. used 23 known racemic drugs to detect 8 types of polysaccharide-coated CSPs and 2 types of immobilized polysaccharide CSPs using three different co-solvents and obtained these results using SFC for the first time. The elution characteristics of polysaccharide-based phospholipids show that the elution performances of phospholipids using different CSPs and co-solvents vary widely [114]. Enantiomeric resolution of a limonenyl bicyclic 1,3-amino alcohol and 1,3,5- and 1,3,6-aminodiols on CSPs based on polysaccharides by the LC NP and SFC methods has been reported, and the effects of mobile phase composition, column temperature, and structures of analytes and CSs on retention and selectivity were investigated for these systems. Hydrogen bonding and π-π interactions affected the chiral identification of the analyte. In most cases, the identification of enantiomers is controlled control by enthalpy. In SFC, only the structure of the polysaccharide backbone affects the elution order [115]. Zhao et al. determined that the modifier pH and column temperature were important factors for the chiral separation of SFC for the separation of chiral perfluorooctane sulfonate. They separation of perfluoro-X-methylheptane sulfonate (X m-PFOS) by SFC, where X m- represents the carbon position of the branched CF3 group. This article also indicating that the SFC method is more economical and environmentally friendly than HPLC (Figure 6a,b) [116]. By using a variety of mobile phases, the ability of HPLC and SFC to separate enantiomers is widely compared. We can more directly see the advantages of SFC compared to HPLC (Figure 6c) [112]. SFC is not as extensive as LC, but its advantages are also obvious. More developments may be worthy of discussion.

### 2.5. High-Speed Countercurrent Chromatography

HSCCC is a chromatographic separation technology developed based on HPLC. It is a liquid–liquid chromatographic separation technology, which means that the mobile phase and the stationary phase are both liquids. The advantage is that there is no irreversible adsorption, no sample loss, no pollution, high efficiency, and fast speed. When two incompatible phases are added to the column, chiral compounds are separated based on their distribution coefficients in the two phases [117]. At present, the most commonly used two-phase recognition in HSCCC occurs when different chiral reagents are added to the stationary phase and mobile phase to study the separation of chiral compounds. When a CS is added to the mobile phase, competitive adsorption with the enantiomer will occur, which will significantly change the intermolecular attraction between the chiral substance and the stationary phase CS through its auxiliary recognition ability. Increasing the selectivity of the stationary phase CS for the chiral compound increases the separation selectivity coefficient and the resolution [118]. Generally, compared with other traditional chromatographic separation technologies, HSCCC has the advantages of a wide application range, large economic benefits, fast separation rate, and simple practical operation [119,120]. In addition, studies have shown that the two phases of a weakly polar solvent system are usually composed of *n*-hexane and water. Methanol, ethanol, and ethyl acetate can be added to adjust the polarity of the solvent system. The two phases of the medium polarity solvent system are composed of chloroform and water. Methanol, ethanol and ethyl acetate can be added to adjust the polarity of the solvent system. The two phases of a strong polar solvent system are usually composed of n-butanol and water. Methanol, ethanol, and ethyl acetate can be added to adjust the polarity of the solvent system or adjust the pH to adjust the polarity. In theory, the choice of solvent system should first involve selecting the best solvent based on the physical and chemical properties of the sample and then selecting the corresponding solvents to obtain the most selective multisolvent system [121]. Among CSs, l-proline and CD are widely used in HSCCC chiral separation, and the technology is relatively mature. Other CSs derived from HPLC technology are also sometimes used in HSCCC technology. For example, Qiu [122] and others used sulfobutyl ether-β-CD as a CS to successfully separate enantiomers of acetyltropic acid in the chiral separation of HSCCC. Wang [123] and others successfully combined Cu(II)-[1-butyl-3-methylimidazolium][l-Pro] (Cu(II)-[BMIm][l-Pro]) with hydroxypropyl-β- CD (HP-β-CD) for use in HSCCC chiral systems. The dual CSs successfully isolated an intractable naringenin (NRG) racemic mixture with 98% purity from samples under the best conditions, which showed the excellent performance of this kind of CS in chiral separation. In addition, this work clarified the chiral recognition mechanism of this type of dual CS agent. Specifically, Cu(II) interacts with the 5-OH and 2-carbon alkyl groups of NRG and then forms a ternary complex with the [BMIm] [l-Pro] complex. Then, the alkyl tail of [BMIm] [l-Pro] is in close contact with HP-β-CD. Based on the spatial phases of NRG and [l-Pro], the diastereomeric quaternary ammonium salt complex of (-)-NRG is more stable than (+)-NRG. Therefore, using HP-β-CD and Cu(II)-[BMIm] [l-Pro] as CSs in the stationary phase retains (-)-NRG more effectively than other methods. Thus, the formation of Cu(II)-[BMIm] [l-Pro], HP-β-CD and NRG quaternary ammonium complexes is the basis for this successful resolution (Figure 7a). Han [119] developed a biphasic chiral recognition system using Cu(II)-*N*-n-dodecyl-l-proline and HP-β-CD as additives to separate the enantiomers of aromatic AAs by HSCCC. The effects of CS concentration and enantiomeric concentration on the separation parameters of chiral compounds were systematically studied (Figure 7b,c). The α values of chiral compound 1 (Figure 7b) and chiral compound 2 (Figure 7c) all increase with the increasing *N*-n-dodecyl-l-proline and HP-β-CD concentrations and show an upward trend, reflecting the synergy of the two CSs. Conversely, as the enantiomer concentration increased, the R values of all compounds decreased significantly. At a concentration of 1.5 mmol/L, the baseline separation was approximately 1.0. After the best separation parameters were obtained, (*R*,*S*)-phenylalanine and (*R*,*S*)-3,4-dimethoxyphenylalanine were successfully separated, allowing the mechanism of coordinated enantiomer recognition based on Cu(II)-*N*-n-dodecyl-l-proline and HP-β-CD to be discussed. In addition, in recent years, the number of reports on HSCCC technology classified by isomer molecular type has increased. For example, Park [124] and others used n-butanol/isopropanol/water (5:1:6, *v*/*v*/*v*) as the solvent system for countercurrent separation, and different concentrations of formic acid solutions were added to successfully isolate and purify sennoside A, A1, and B from senna leaves (*Cassia acutifolia*). Wang [125] proposed a counter current chromatography method using HP-β-CD as a chiral resolving agent. This method used n-hexane/20–70% methanol containing 50 mmol/L HP-β-CD (1:1, *v*/*v*) as a two-phase solvent system and successfully separated ten epimers of aromatic acid menthol ester.

### 2.6. Thin-Layer Chromatography

TLC is a branch of chromatography that is often used for rapid separation and sensitive detection of substances. It is also one of the most common and simplest chromatography techniques in chiral separations. In contrast, in the separation of chiral compounds, TLC has a wide range of options, easy adjustment of chromatographic parameters, simple equipment, easy operation, and intuitive results. In addition, the mobile phase can be quickly changed in TLC. Chiral compounds have good development prospects [126,127]. Chiral thin-layer boards commonly used in TLC are mainly cellulose and its derivatives and are sometimes impregnated with chiral selective agents. Among them, the most common thin-layer sheet impregnated with a CS is a copper (II) composite thin-layer sheet with an α-AA alkyl derivative. In addition to copper (II), zinc (II), mercury (II), and silica gel impregnated with α-chitin and α-chitosan can be used to isolate AAs and their derivatives. Chiral TLC plates impregnated with a CS are currently an effective way to directly separate chiral compounds [128]. In recent years, some researchers have applied TLC to the separation of chiral compounds. For example, Manisha [129] et al. used four l-AAs as chiral additives and performed TLC on silica gel plates to split (RS)-ketorolac. Malik [130] and others used bovine serum albumin (BSA) as a stationary phase and a simple organic solvent without buffers or inorganic ions to achieve chiral separation of five analytes. Later, Malik [131] and others reported that ligand exchange TLC directly separated (RS)-ketorolac and (RS)-etodolac with Cu^2+^ in three l-AA complex ions as chiral additives. To pursue the suggestion of Schmid et al., chiral ligand exchange chromatography (CLEC) was used to form a ternary mixed metal complex between a complex metal ion and a chiral compound. Two of the AA molecules acted as chelating ligands for Cu^2+^, and the competing enantiomer of the enantiomeric mixture replaced one chelating AA molecule during the separation process (Figure 8a). Therefore, a diastereomer complex of the two enantiomers was formed to separate these enantiomers, and then through the interaction of normal silica gel in TLC, different stabilities and different retentions achieved good separation results. In the latter stage, iodine vapor was used for spot visualization, and the two separation results were compared under the best experimental condition. Wang [132] et al. reported the study of MIPs as CSPs for TLC. This study used d-naproxen as a template and polymerized it in an acetonitrile/methanol (AM) mixed solvent. MIPs with particle sizes between 10 and 90 μm were prepared. Through UV-vis spectroscopy, it was found that with increasing AM concentration, the absorption wavelength of d-naproxen showed a redshift in the range of 220 nm to 270 nm, and the absorption intensity decreased significantly (Figure 8b). This shows that there was some intermolecular interaction between AM and naproxen. Since naproxen has methoxy groups and carbonyl groups and acrylamide belongs to AAs with amide groups, it was presumed that the key interaction was hydrogen bonding. Additionally, based on the molecular self-assembly process, the distribution of large pi (π) bonds and electron clouds changed, and the synthetic route and recognition principle of MIP were deduced (Figure 8c). Later, the effect of acetic acid content on the separation of chiral compounds was studied, and it was found that when the acetic acid content was 4%, the chiral α was 1.58. Khatri [133] et al. cross-linked polyvinyl alcohol with glutaraldehyde and β-CD to form β-CD-PVA-GA membrane, which is used as CS in TLC, and the membrane has excellent enantiomeric separation performance.

## 3. Chiral Membrane Separation Technology

Membrane-mediated chiral separation is a new technology with broad application prospects, and it has received more and more attention in recent years. Membrane-based chiral separation methods have unique advantages compared to traditional chiral chromatographic separation methods, such as low cost and energy consumption, simple equipment, sustainable operation, high environmental protection, and easy realization of large-scale industrial production and many more [134,135,136,137]. The common preparation method of chiral separation membrane is base membrane modification and modification method, that is, by coating, dipping, grafting, etc., the material with chiral recognition function is loaded on the support layer based on polysulfone, polyamide, cellulose, and other materials. In order to further improve the ease of preparation of chiral separation membranes, the direct film formation method has gradually attracted the attention of researchers. This method mainly uses materials with chiral recognition functions such as cyclodextrin, chitin, chitosan and other natural substances to directly form into films. The disadvantage is that this method has low universality for most chiral compounds. In addition, membrane technology can be combined with molecular imprinting technology to design molecular imprinted membrane. The molecular imprinted membrane not only has specific recognition characteristics, but also can realize continuous operation and effective separation of substances with specific structures [138]. Based on the above-mentioned different preparation methods, as a new hot spot in the field of chiral separation research, chiral membrane resolution technology has gradually shifted the research focus to the preparation of chiral separation membranes that are easy to industrialize or target. In terms of state, chiral separation membranes include chiral liquid membranes and chiral solid membranes.

### 3.1. Liquid Membrane Separation

Liquid membrane separations were proposed and patented by Dr. Li in 1968. The separation process is similar to the extraction process. The solute first enters the membrane phase from the liquid phase, and it is transferred to the membrane phase by diffusion. The other side of the membrane phase is back-extracted to the receiving phase, that is, extraction and back-extraction during liquid membrane separation are performed simultaneously at the interfaces on both sides of the membrane. Based on this, chiral liquid membrane is a combination of chiral extraction technology and membrane separation. Generally, when a liquid membrane separates a chiral substance, the chiral selector dissolved at the interface between the organic solvent and the donor phase will form a complex with the target enantiomer, and then the difference in the concentration of the chiral compound between the two sides of the membrane will act as the driving force to achieve chirality material transfer. In addition, because the liquid membrane has a stronger affinity for a specific chiral compound than for other chiral compounds, this selection principle is used for extraction and resolution [139].

At present, according to different structural properties and preparation processes, chiral split liquid membranes can be divided into bulk liquid membranes (BLMs), emulsified liquid membranes (ELMs), and support liquid membranes (SLMs). BLM is relatively stable, easy to configure, and the easiest liquid membrane to operate. The membrane phase does not need to be supported, and the material liquid phase and the receiving phase are separated by only a relatively thick layer of immiscible organic phase fluid. Recently, four new enantiopure crown ethers containing a diphenylphosphinic acid unit were synthesized [140], and these units exist as ligands in an aqueous source phase/lipophilic organic bulk liquid membrane that is continuously controlled by the pH of the medium. In the aqueous receiving phase system, protonated phenylethylamine and phenylglycol are transported enantioselectively. Compared with the BLM, the ELM has a higher mass transfer rate and good separation performance. The oil phase and the water phase containing the film solvent are made into a *W*/*O* emulsion under high-speed stirring, and this emulsion is dispersed to another *W*/*O*/*W* emulsion formed in this kind of aqueous solution [141]. However, due to the cumbersome operation of the emulsion membrane system, the small range of application and membrane leakage, it has not been applied in the separation of chiral compounds until recent years. In addition, due to its unique advantages of high selectivity and high throughput, SLM has attracted more and more attention in recent years [142]. They are also the most widely used liquid membranes in the separation of chiral compounds. The membrane is usually a porous inert base membrane (support) immersed in a membrane solvent in which a carrier is dissolved, and through the action of surface tension, the membrane solvent is filled with micropores to form the SLM [143]. During use, the liquid film is lost after contacting the liquid phase and the receiving phase, which causes the function to decline. Therefore, the support material has a great impact on the separation process. Generally, hydrophobic materials such as polyethylene, polypropylene, or polytetrafluoroethylene are used. Porous membranes work well as support materials. For example, Zhang [144] and others have studied a chiral liquid membrane using l-tartaric ester dissolved in n-octane as the liquid membrane phase and polyvinylidene fluoride hollow fibers as the membrane carrier to separate racemic ibuprofen. In this work, the hollow fiber module was operated in recirculation mode (Figure 9a), the ibuprofen buffer was dissolved in sodium phosphate–phosphate and passed through the flow tube side, and the sodium phosphate–phosphate buffer solution was passed through the shell side. Another study [145] used a hollow fiber support liquid membrane for chiral separation of salbutamol. In this study, O,O-dibenzoyl-(2S,3S)-tartaric acid (+)-DBTA and di(2-ethylhexyl)phosphoric acid were used as chiral and achiral extractants, respectively. The separation of enantiomers was achieved by simultaneous extraction and stripping. Using CS Chem3D Ultra, the initial structures of (+)-DBTA and salbutamol enantiomers were constructed. At the same time, a density functional theory method was used to optimize each structure. No symmetric constraints were found. When the bonds were calculated after a long time, the optimized complex configuration was obtained. At the same time, chiral recognition was achieved through hydrogen-bond interactions. The number of hydrogen bonds between S-salbutamol and (+)-DBTA (Figure 9b) was less than the number of hydrogen bonds between R-salbutamol and (+)-DBTA (Figure 9c); that is, the complex of (+)-DBTA and R-salbutamol was more stable. Different from the previous research, Gaálová et al. [146] recently successfully explored the formation of a non-porous composite membrane for the first time. This study dissolves piperazine or m-phenylene diamine (MPD) in deionized water to form an aqueous phase and dissolves 1,3,5-trimesoyl chloride (TMC) in n-hexane to form an organic phase. Then, the chiral substance (S, S)-1,2 diaminocyclohexane (DACH) was used to replace the variable part of MPD, and finally, a membrane suitable for enantiomeric separation was obtained. At the same time, the three-point interaction model was used to study the mechanism by which TMC + MPD/DACH recognizes Trp enantiomers. It was found that the combination of l-Trp with TMC + MPD/DACH conforms to the proposed mechanism and that the four proposed interactions are all involved (Figure 9d), but d-Trp maintains only two hydrogen bonds and loses the π-π stack (Figure 9e), which is the arrangement required based on its configuration.

The use of liquid chiral membranes is a promising method to expand the separation of enantiomers. It has been applied to the separation of AAs, drugs and amines, or in combination with biotransformation. Although different types of chiral liquid membranes and multiple applications have been described, their actual scope of application still has limitations. For the purpose of expanding the wide range of applications, it is necessary to research and develop some new materials including chiral inorganic structures.

### 3.2. Solid Membrane Separation

In 1986, the first work on chiral solid membranes was proposed by Y. Osada. This research fixed l-menthol or d-camphor to the surface of polymer-based membranes using plasma technology. The separation medium was an aqueous solution, and pressure drove the resolution of chiral samples of tryptophan (Trp) or proline and exploration of the chiral solid membrane. Compared with chiral liquid membranes lacking long-term stability, chiral solid membranes have better stability, permeability, and chiral selectivity, and can adapt to large-scale industrial production [137,138]. Therefore, the current research on chiral films mainly focuses on solid films. Currently, chiral two-component solid membranes can be categorized as bulk solid membranes (BSMs), modified solid membranes (MSMs), and molecularly imprinted solid membranes (MISMs) based on the membrane properties and manufacturing processes. Usually, a chiral base membrane material is used to prepare a cast film under certain conditions, and then the membrane structure is fixed by cross-linking and other methods, and finally BSM that can realize the separation of chiral compounds is obtained [141]. For example, in 2014, Lei et al. [147] first prepared a methyl methacrylate-*N*-isopropylacrylamide (MMA-NIPAm) polymer with surface functionalization and monodispersity through atom transfer radical precipitation polymerization. The resulting chiral selective MMA-NIPAm cation exchange membrane can realize the efficient separation of racemic equol. In addition, research has been conducted to prepare a chiral polymer membrane with 64–78% amino acid separation rate by in-situ interfacial copolymerization of trimesoyl chloride and l-arginine on a polysulfone ultrafiltration membrane. Unlike BSM, MSM uses techniques such as covalent grafting, dipping, or esterification to fix and coat the surface or pores of the base film with a polymer layer with chiral separation, such as a polymer containing a chiral selective agent, polymers of sugar side chains or chiral metal complexes and polymers of acetylated β-cyclodextrin surface-functionalized cellulose are used to prepare composite membranes that can achieve efficient separation of chiral compounds [148]. That is, the chiral separation at this time mainly occurs in the surface layer, and the base film plays only a supporting role. At present, the most widely selected materials suitable for solid film modification are polymer surfaces. In recent years, based on the superior mechanical strength, good thermal stability, excellent atomic thickness and chemical inertness of graphene materials, an example of the evolution of a graphene oxide (GO) base membrane into a new type of screening membrane has appeared. For example, in 2017, Meng and his colleagues [149] used l-glutamic acid (Glu) as a chiral selector to modify graphene sheets to facilitate the penetration of target chiral compounds, and then Glu-GO membrane was obtained by vacuum filtration through cellulose acetate membrane acting as a porous carrier. As a permeation model, the study also tested the chiral resolution of 3,4-dihydroxy-D,l-phenylalanine by Glu-GO membrane, and found that it not only has higher selectivity but also has a flux that is 1–2 orders of magnitude higher than that of ordinary chiral separation membrane. In order to achieve large-scale production of chiral separation membranes, a study used [150] mussel-inspired chemistry to prepare chiral separation polysulfone membranes and then modified and covered with dopamine, and then β-cyclodextrin, which acts as a chiral selector, forms a coating on the surface of polydopamine, and the obtained mussel-inspired chemically modified membrane has the same mechanical properties as the normal polysulfone base membrane, and under the optimal pH value, the racemic mixture of tryptophan can be efficiently separated. BSMs are prepared by using a chiral basic film material under certain conditions to prepare a casting film, and then the membrane structure is fixed by crosslinking and other methods to finally obtain a solid film that can achieve separation of chiral compounds [141]. Unlike BSMs, MSMs are coated with a polymer layer with chiral separation on the surface of the base membrane by using techniques such as grafting to prepare a composite membrane that achieves chiral resolution. That is, the chiral separation at this time mainly occurs in the surface layer, and the base film plays only a supporting role. In addition, molecular imprinting can be used to create molecular recognition sites [151]. By introducing molecular recognition sites in the desired template, highly heterogeneous holes can be formed in a solid film. These holes target only one of the enantiomers (D or L), conforming to its configuration to allow it to pass, so the other enantiomer is intercepted to achieve chiral separation. Due to the advantages of being well designed, well recognizable, and practical, MISMs have also become a research hotspot in recent years. In one study [148], a monodispersed methyl methacrylate–*N*-isopropyl acrylamide (MMA-NIPAm) polymer was prepared by a one-pot method and then 3 weight% (wt%) MMA-NIPAm copolymer was dissolved in dimethylformamide solution and cast. Finally, the solvent was completely evaporated on a glass plate under a vacuum at 100 °C to prepare an MMA-NIPAm chiral separation membrane that effectively separated racemic equol. In another study [152], alumina templates coated with cellulose acetate (CA) were used for the analysis and preparation of various membranes. The optimal amount of CA added to the casting solution was investigated. At 10 wt%, the chiral recognition sites after film formation were limited in number, affecting separation (Figure 10a). When the CA content was 15 wt%, the distance between the two chiral sites was appropriate, the feed liquid molecules were sequentially identified according to each chiral identification site, and the separation was good (Figure 10b). However, when the CA content in the casting solution was 20 wt%, more chiral recognition sites appeared in the membrane, which lowered the separation selectivity (Figure 10c). Therefore, a CA content of 15 wt% was selected in the production of the resulting film. Then, using (S)-(+)-mandelic acid as the imprinted molecule, molecularly imprinted nanochannel membranes that had higher permeability selectivity than traditional membranes were prepared.

Regarding solid membranes, MSM usually has higher specificity and selectivity in the interaction between enantiomers and chiral recognition sites, and has good efficiency in separation applications, it is expected to become one of the most promising technologies to achieve large-scale chiral resolution. According to current reports, the separation applications of chiral solid membranes are mostly suitable for AAs. Therefore, polymers such as cellulose derivatives have been developed to construct MSM to realize the wide applicability of various chiral substances and gradually meet the needs of industrial applications, it is also a very interesting area.

## 4. Concluding Remarks and Future Prospects

We summarized many publications about CSPs and membranes and their applications to chiral separation, and the molecular mechanism underlying chiral recognition of enantiomers was further discussed. The choice of the optimum CSP with high selectivity and specificity is the key attribute in the development of enantiomer separations in HPLC. The mechanism of analytical enantioseparations is based on hydrogen bonds, ionic bonds, ion–dipole and dipole–dipole interactions, and van der Waals forces. Chromatographic techniques, especially HPLC, are routinely applied to enantiomer separation, but the sensitivity of enantiomer recognition is still a challenge in chiral resolution. Further study indicated that chiral recognition is not only based on the structures of the selectors and the analytes but also affected by the mobile phase, the background electrolyte, and other solvents in chromatography [76]. Further analysis of the chiral recognition mechanism for predicting chiral separations may modulate analyte-selector interactions in molecular structures and molecular modelling studies coupled with solvents.

In addition, the chiral monolithic chromatographic columns widely used in GC and CE facilitate the development of enantioseparation based on chromatographic technology. HSCCC has the advantages of a wide application range, large economic benefits, fast separation rates, and simple practical operation. Compared with other chromatographic technologies, the SFC system has presented promising potential with the advantages of a short analysis time, high resolution for most analytes, and low consumption of mobile phase components. With regard to the development of CSPs with chromatography technology, HSCCC and SFC are relatively newer technologies than HPLC, GC and CE and appear to be generally promising. Mechanistic studies of CSs are still needed to improve the specificity and selectivity of chromatography technology. Recently, an increasing number of manually modified CSs, such as polysaccharide derivatives and CDs, have been widely used for the chiral resolution of diverse analytes with the advantages of easy preparation and high selectivity.

Furthermore, intrinsic features of membrane-based chiral resolution can be widely used in large-scale industrial applications with the advantages of cost, efficiency, energy savings, ease of scaling up, and continuous operation. Membrane separation methods are promising for chiral resolution with obvious advantages and do not require expensive large-scale instruments or professional operators, unlike chromatographic technology.

## Figures and Tables

**Figure 1 molecules-26-01145-f001:**
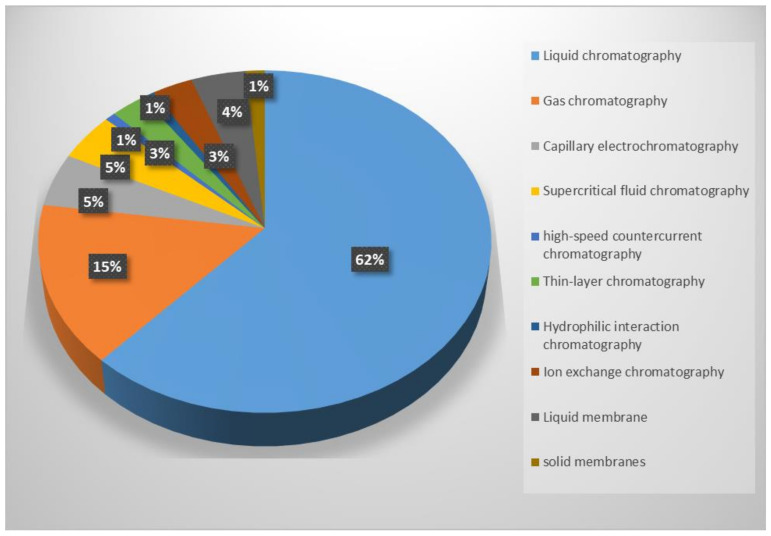
Publications for separation technology classification, including the chromatographic separation methods of liquid chromatography, gas chromatography, capillary electrochromatography, supercritical fluid chromatography, high-speed countercurrent chromatography, thin-layer chromatography, and hydrophilic interaction chromatography and chiral membrane separation methods with liquid membranes and solid membranes. Source: Web of Science.

**Figure 2 molecules-26-01145-f002:**
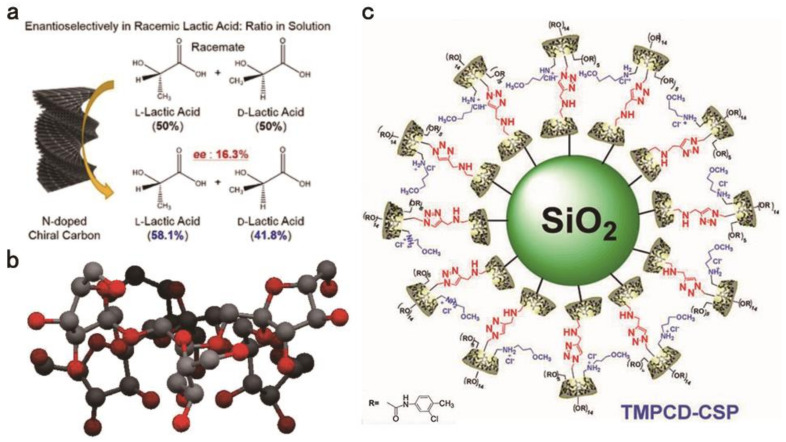
Chiral selector materials for HPLC. (**a**) Enantioselective adsorption of lactic acid by chiral nitrogen-doped carbon-sheath nanofibers and ee of the product solution after incubation. (**b**) Side view of the cyclofructan-6 crystal structure; black indicates carbon atoms, and red indicates oxygen atoms. (**c**) Structure of a cationic CD clicked bilayer CSP.

**Figure 3 molecules-26-01145-f003:**
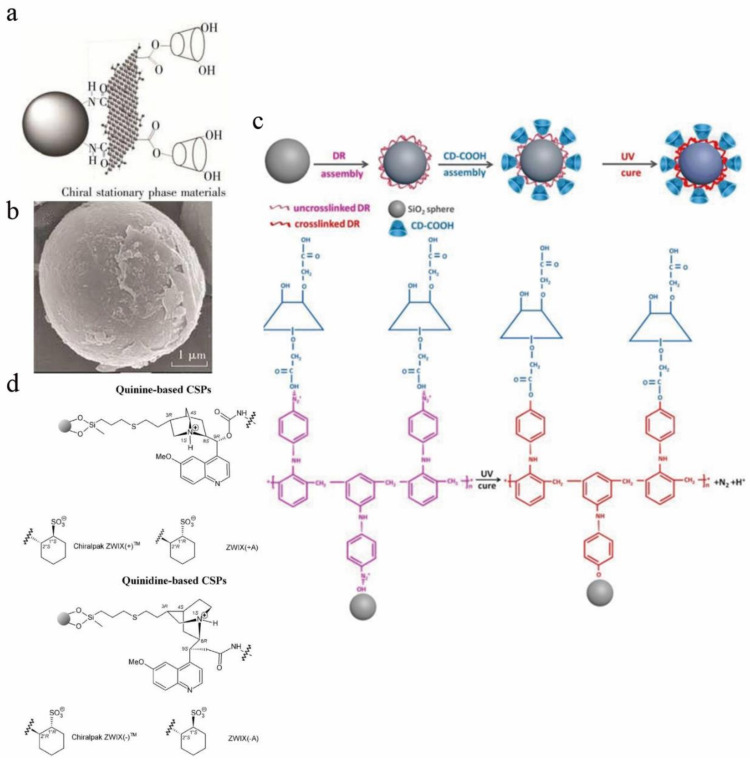
Chiral selector materials for HILIC and ion exchange chromatography. (**a**) Structure of β-CD-GO@SiO_2_. The β-CD-GO@SiO_2_ was synthesized by bonding graphene oxide (GO) to aminosilica (NH_2_-SiO_2_) and then bonding β-CD. (**b**) Scanning electron micrograph of β-CD-GO@SiO_2_. (**c**) Synthesis process of light-assisted preparation of cyclodextrin-based CSPs. (**d**) The structure of zwitterionic CSPs that has been successfully commercialized. The commercial product name is Chiralpak ZWIX(+)^TM^/Chiralpak ZWIX(−)^TM^.

**Figure 4 molecules-26-01145-f004:**
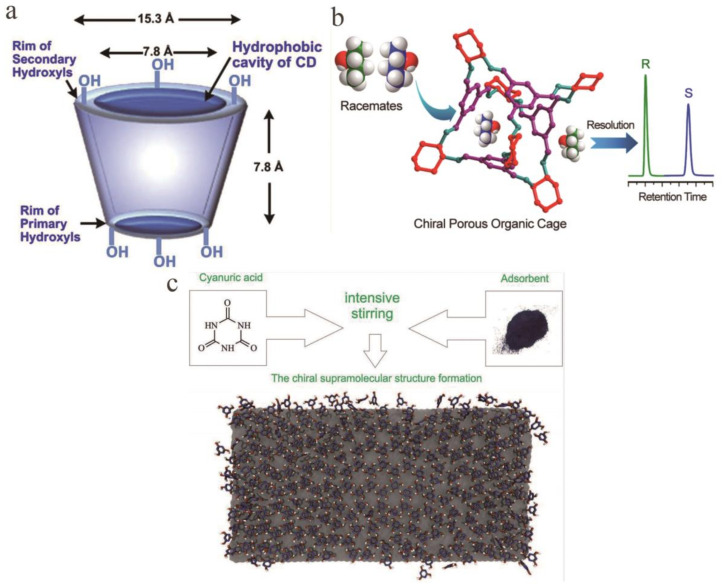
Chiral selector materials for gas chromatography (GC). (**a**) Truncated cone structure of cyclodextrin showing the hydrophilicity of the outside of cyclodextrin due to the distribution of hydroxyl groups there. (**b**) Application of homochiral porous organic cage to GC to separate enantiomers. (**c**) Principal scheme of cyanuric acid chiral superstructure formation on the graphene surface.

**Figure 5 molecules-26-01145-f005:**
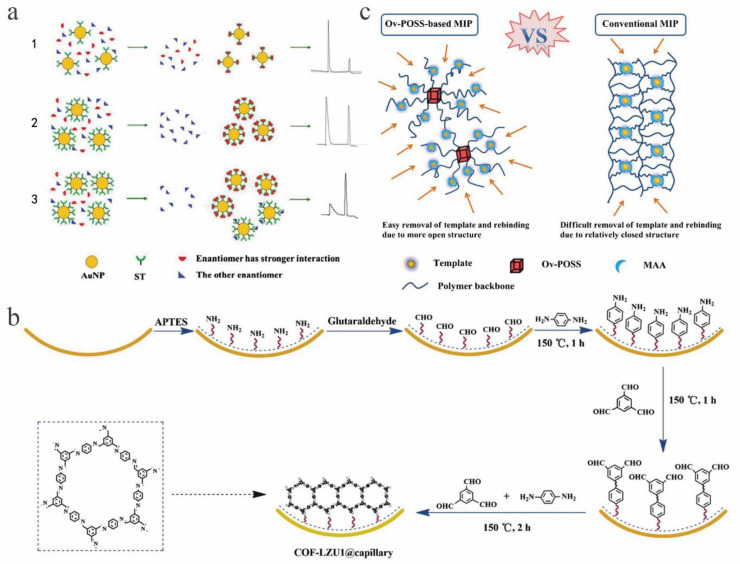
Chiral selector material for capillary electrochromatography (CEC). (**a**) Schematic diagram of chiral separation in streptomycin-modified gold nanoparticle (ST-AuNP) systems with different concentrations. (1) At low ST-AuNP concentration, (2) with the ST-AuNP concentration increasing up to the optimal value, (3) at an ST-AuNP concentration that is too high. (**b**) Scheme for the growth of COF-LZU1 on the inner wall of aldehyde group-functionalized capillary. (**c**) Schematic representation of Ov (octavinyl)-based molecularly imprinted polymers (MIPs) and conventional MIPs.

**Figure 6 molecules-26-01145-f006:**
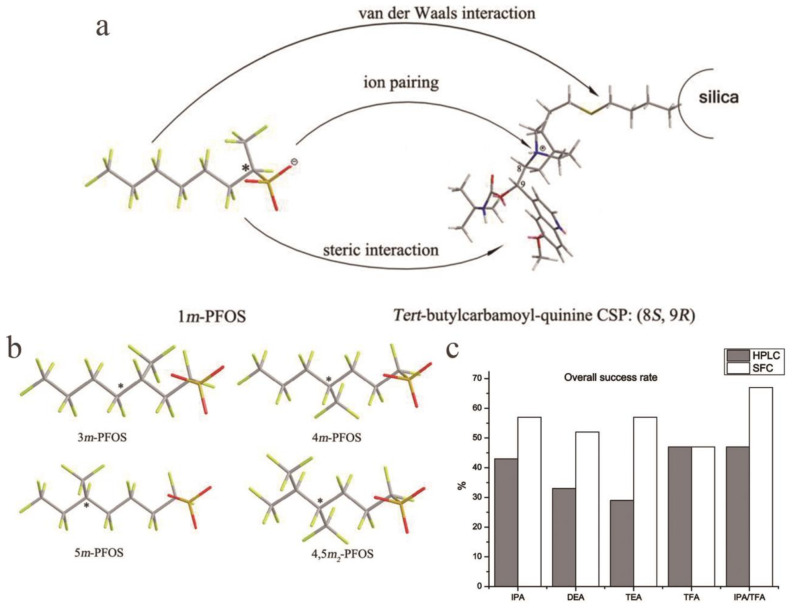
Chiral selector materials for supercritical fluid chromatography (SFC) and the separation ability of SFC and HPLC under the same mobile phase ratio. (**a**) Possible intermolecular interaction between 1m-PFOS (perfluoro-1-methylheptane sulfonate) and a tertbutyl carbamoylquinine-based weak anion exchange chiral stationary phase (Chiralpak QN-AX). (**b**) Structures of 3*m*-PFOS, 4*m*-PFOS, 5*m*-PFOS and 4,5*m*-PFOS. (**c**) The effects of high-performance liquid chromatography and supercritical fluid chromatography on the separation performance were compared. Mobile phase is hex (n-hexane)/PrOH (propan-2-ol)/X or CO_2_/PrOH/X = 80:20:0.1 (*v*/*v*/*v*), where X is IPA (isopropylamine), DEA (diethylamine), TEA (triethylamine), TFA (trifluoroacetic acid), or a 1:1 (*v*/*v*) combination of IPA:TFA.

**Figure 7 molecules-26-01145-f007:**
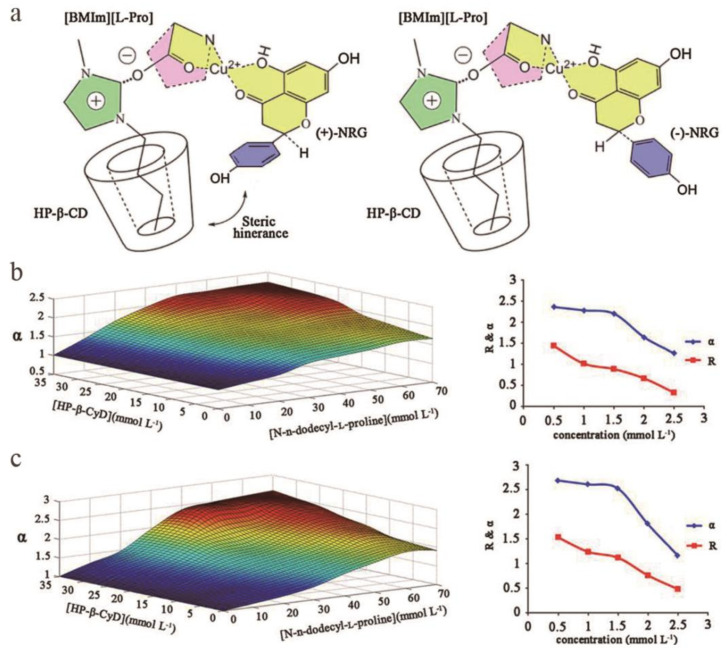
Chiral selector materials for high-speed countercurrent chromatography (HSCCC) and optimization of separation conditions. (**a**) Geometries of quaternary complexes containing HP-β-CD, [BMIm][l-Pro], Cu(II), and NRG enantiomers. (**b**) Effects of the main factors influencing α and R of complexes 1. (Left) concentration of chiral selector and (right) concentrations of enantiomers. Solvent system: n-butanol/0.1 mol/L phosphate buffer solution (1:1, *v*/*v*) at pH 5.50 and 10 °C. (**c**) Effects of the main factors influencing α and R of complexes 2. (Left) concentration of chiral selector and (right) concentrations of enantiomers. Solvent system: n-butanol/0.1 mol/L phosphate buffer solution (1:1, *v*/*v*) at pH 5.50 and 10 °C.

**Figure 8 molecules-26-01145-f008:**
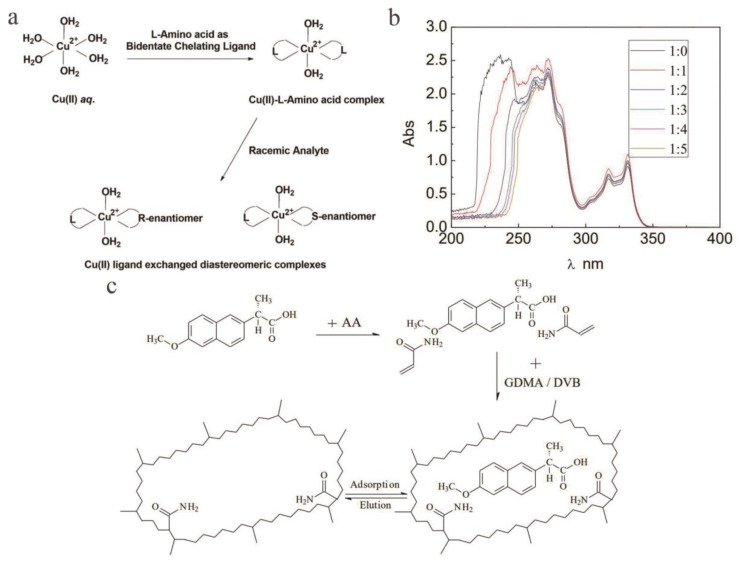
Chiral selector materials and selection recognition mechanism for thin-layer chromatography (TLC). (**a**) Formation and structures of a ternary complex and a ligand-exchanged complex on the resolution of racemic analytes. Cu(II) aq. represents the complexed state of Cu(II) in aq. medium. (**b**) UV-vis spectra of naproxen samples with different molar ratios of naproxen to acetonitrile/methanol (AM). (**c**) Synthesis route for the imprinted polymer and its recognition mechanism.

**Figure 9 molecules-26-01145-f009:**
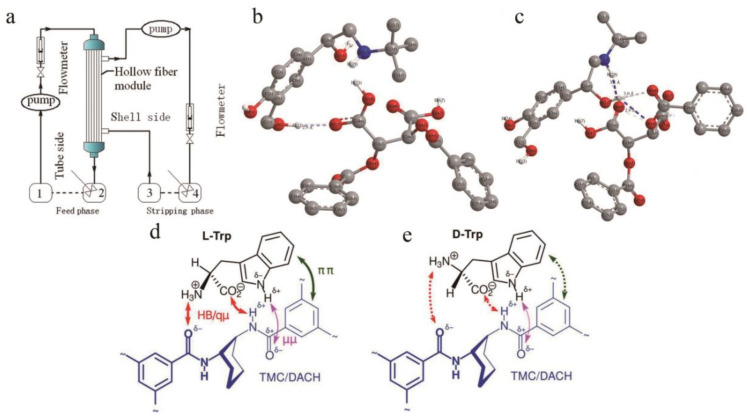
Flow diagram of chiral liquid membrane separation and schematic diagram of the interaction between (+)-DBTA and target. (**a**) Flow diagram of enantioselective separation of racemic ibuprofen by chiral liquid membrane. (**b**) Optimized geometry of the complex between (+)-DBTA and S-salbutamol. (**c**) Optimized geometry of the complex between (+)-DBTA and R-salbutamol. (**d**) Binding of l-Trp to a selector. (**e**) Binding of d-Trp to a selector. The proposed chiral recognition mechanism of Trp in the TMC+MPD/DACH polyamide active layer. Trimesoyl amide groups depicted with thinner bonds are farther from the observer, behind the Trp molecule in its binding position. Arrows indicate interactions assumed within the three-point model: Red (HB/qµ)—strong H-bonding and/or ion-dipole interactions, green (ππ)—a π–π stacking interaction, and magenta (µµ)—an “auxiliary” dipole–dipole interaction in antiparallel alignment.

**Figure 10 molecules-26-01145-f010:**
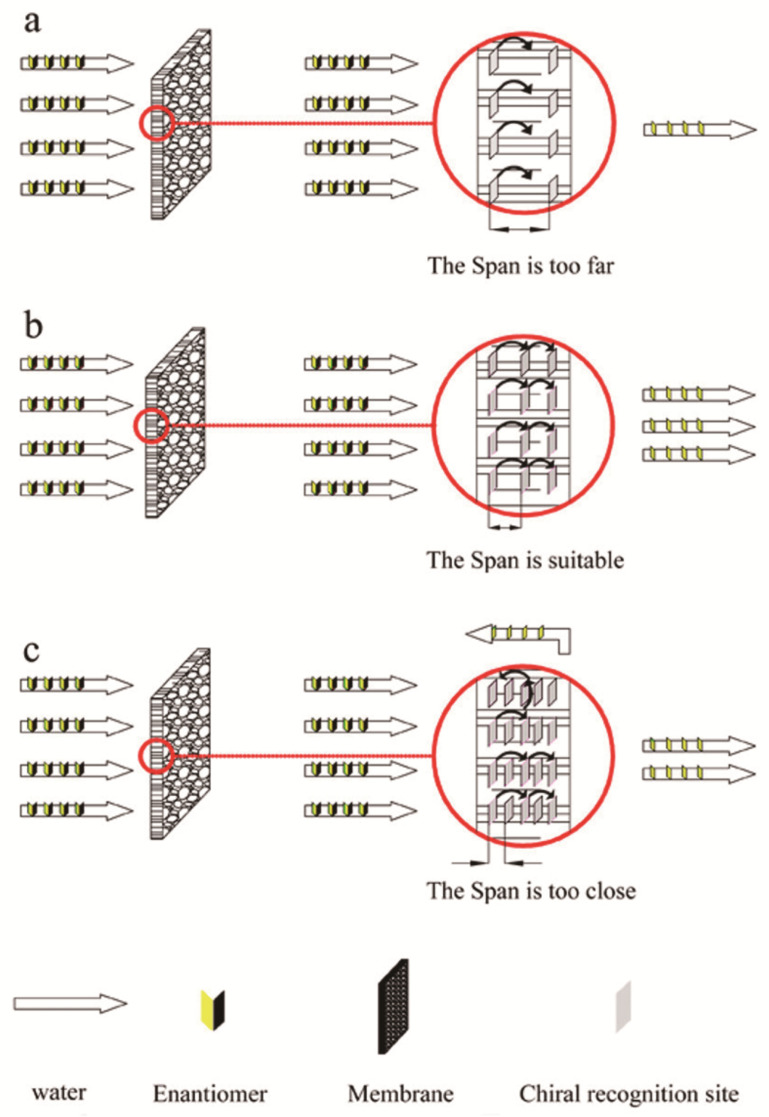
Chiral recognition site schematics of the membrane. (**a**–**c**) represent cellulose acetate (CA) membranes prepared with 10, 15, and 20 wt% CA, respectively. (The small grey diamonds represent chiral recognition sites, and the denser the diamonds are, the more chiral recognition sites on the membrane).

**Table 1 molecules-26-01145-t001:** Target characteristics and commercial columns corresponding to chiral stationary phases (CSPs).

Types of CSPs	Basic Material	Target Characteristics	Commercial Column
Polysaccharides	Amylose or cellulose	Widely applicable, such as compounds containing amide groups, aromatic ring substituents, carbonyl nitro, sulfonyl, cyano, hydroxyl, amino and other groups, and amino derivatives	Chiralcel^®^OD, Chiralpak^®^IB, Lux^®^Amylose-1, Lux^®^i-Amylose-1
Cyclodextrins	β-cyclodextrin	Widely applicable, such as hydrocarbon compounds, sterols, phenol esters and their derivatives, aromatic amines, polyheterocycles	B-DEXTM225, Astec Cyclobond^®^, I 2000 RSP, LiChroCART^®^250-4, ChiraDex^®^
Proteins	Enzymes, plasma proteins, receptor proteins	Water-soluble medicine	Chiralpak^®^HAS, Resolvosil^®^BSA, Chiralpak^®^AGP
Crown ethers	Macrocyclic polyether	Amino acids, amino alcohols, primary amines	Crownpak^®^ CR(+)/CR(−), Chirosil^®^ RCA(+)/RAC(−)
Pirkle type	Amine, amino alcohol, amino acid derivative Compounds, anthrone derivatives	Widely applicable, CSPs designed by analyzing the target	Whelk-O1^®^, ULMO^®^, Chirex^®^
Ion exchange type	Cinchona alkaloids, sulfamic acid	*N*-protected amino acid, *N*-protected amino group, sulfamic acid, amino phosphoric acid, aromatic carbonyl acid	Chiralpak^®^QN-AX, Chiralpak^®^QD-AX, Chiralpak^®^ZWIX(+), Chiralpak^®^ZWIX(−)
Macrocyclic glycopeptides	Avomycin, Ristomycin A, Vancomycin, Teicoplanin and Teicoplanin aglycone	Widely applicable, such as amino acids, peptides, non-steroidal anti-inflammatory drugs	Astec^®^ CHIROBIOTIC^®^ V, Astec^®^ CHIROBIOTIC^®^ R, Astec^®^ CHIROBIOTIC^®^ TAG
Cyclofructans	Cyclic oligosaccharides	Primary amine, acid, secondary amine, tertiary amine, alcohol	Larihc^®^ CF6-RN
Porous organic materials	Covalent organic framework, metal organic framework, metal organic cage, mesoporous silica	Halogenated hydrocarbons, ketones, esters, ethers, organic acids, alkylene oxides, alcohols and sulfoxides	/

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
