# Peer review of "Chiral Recognition for Chromatography and Membrane-Based Separations: Recent Developments and Future Prospects"

_molecules, 2021, doi:10.3390/molecules26041145_

Round 1

Reviewer 1 Report

  • Unfortunately, authors did not mark any changes in the initial version of the manuscript with any new print font or color. So that it is not possible to check any improvements and makes it necessary to read the whole text once again. My basic impressions of the first variant, formulated in my first report, remain basically valid for the revised version.
  •   
  • The review is prepared by several persons and, therefore, the scientific quality of different parts is different. At least, I find it impossible to write in any scientific publication such naïve and unprofessional sentences as “The biggest difference between IEC and HPLC is that it uses different chromatographic columns.” (Line 235).
  • To show some examples of printing and English stile errors, please, pay attention to Lines 39, 245, 294, 313, 318, 462-464.
  •  
  • On the other hand, as I mentioned earlier, the paper brings many recent references in the huge and highly important field of chiral resolutions, so that, combined with full names of the references, the manuscript could by useful for the international researcher community.
  •  
  • In short, I would not object publishing of the submitted material in Molecules.

Author Response

Thank you for your comment, please refer to the attachment for the reply to the comments.

Reviewer 2 Report

In this manuscript, the authors provide a review on chromatographic methods to separate enantiomers. This review might be useful particularly for those looking for new analytical methods to measure or separate small amounts of chiral compounds. As a scientist with a just a tangential interest in this area, this review is not an easy to read from beginning to end. While the writing is reasonable, notwithstanding some suggestions provided in the PDF, the significant use of multiple acronyms throughout the text provided significant speed bumps that slowed the comprehension of the narrative. Moreover, the lack of parsing of thoughts (i.e. paragraphs) in the latter halve of the review, further detracted from the readability of the manuscript. Once these issues are addressed, publication of this review is warranted.

Author Response

(The authors gave the same response as above.)

Round 2

Reviewer 2 Report

The authors made substantial revisions in this resubmission that addressed most of the original concerns. One suggested change is given below. Line 26-27: …Enantiomers are defined as a pair of compounds that are non-superimposable mirror images of each other, and such compounds are usually called chiral molecules.

Author Response

Thank you for your comments, we have revised the relevant content in the manuscript.

This manuscript is a resubmission of an earlier submission. The following is a list of the peer review reports and author responses from that submission.

Round 1

Reviewer 1 Report

 Review  

Chiral recognition for chromatography and membrane-based separations: Recent developments and future prospects  

Yuan Zhao, Xuecheng Zhu, Wei Jiang, Huilin Liu* and Baoguo Sun   

Unfortunately, I cannot recommend accepting the manuscript for publication in a solid international journal as I fail to identify any substantial group of readers who could find the paper informative enough (with the exception of the list of recent references).

Authors tried to achieve an impossible goal – to give an overview of all available methods of chiral separation in a relatively short manuscript. The paper starts with simple basic definitions, which gives the impression of providing a beginner in the field with important hints for selecting an optimal approach to solve their practical problem. But the following text presents a lot of terms, chemical names and abbreviations, meaningless details, etc. However, with only few chemical formula, missing analysis of benefits versus weak points of individual techniques and no recommendations concerning optimal application area of each technique, the potential beginner will learn almost nothing. Here, a correct classification of all approaches and mentioning seminal works of “chiral guru” like Pirkle, Davankov, Schurig, Okamota and others would be more helpful.

A more experienced specialist finds the quality of the most parts of the manuscript to be insufficient. Absolutely impossible sound such statements as, e.g. “The main difference from HPLC is that HSCCC uses a spiral chromatography column.” (line 528 – about counter current chromatography!).

Authors promise describing chiral Hydrophilic interaction chromatography (separately of HPLC – why?), but then forget about it.                                                                                          

“Among them, the better resolution of primary amines in partially derivatizing aliphatic  functional groups may be due to the destruction of the internal hydrogen bonds of cyclofructan and  the relaxation of its internal structure during the derivatization of hydroxyl groups.”   (extremely unclear - lines 125-126).

Mentioning preparation of “novel bilayer cationic CD CSPs through a click reaction” requires a more detailed explanation. (lines 171-175)

“Khatri [133] et al. used β-CD to cross-link glutaraldehyde (GA) into polyvinyl alcohol (PVA) to form a β-CD-PVA-GA film” (Should mean: “to cross-link polyvinyl alcohol with glutardialdehyde and β-CD” - line 624)

There are several chemical errors, as:  Acethyltropic acid was resolved, not acetoacetic acid! [line 547, Ref 122]. “1,3,5-triformyl chloride (TMC)” does not exist (line 711).

I find the rather comprehensive list of recent references most useful for all specialists in the field.

Moreover, I find the last part of the manuscript concerning membrane separation technique to be better than all others. I could recommend extending this single part to a separate review on this rather perspective approach to preparative chiral separations.

Reviewer 2 Report

This review deals on the recent developments in the separation and detection of chiral compounds. It is accurately written and according to my opinion is acceptable for publication as is.

I have only one suggestion. Please, correct the term “chiral enantiomer” because it is redundant.

The article to be evaluated is a review. I find the review in question well written. The topic analysed is undoubtedly interesting as it relates to the state of the art in the field of enantiomeric separation of chiral molecules. Being a review, the manuscript does not present original arguments or conclusions on the arguments presented.

The authors' aim was to present an overview in the field of chiral separation by chromatography or membrane filtration and I think they did it in a clear and readable way. For these reasons I think the article is acceptable for publication.